# Lung Ultrasound as a Triage Method in Primary Care for Patients with Suspected SARS-CoV-2 Pneumonia

**DOI:** 10.3390/jcm11216420

**Published:** 2022-10-29

**Authors:** María Belén Guzmán-García, Alicia Mohedano-Moriano, Jaime González-González, José Manuel Morales-Cano, Ricardo Campo-Linares, César Lozano-Suárez, Tyrone Paul Estrada-Álvarez, Mª Mar Romero-Fernández, Esther Vanesa Aguilar-Galán, Juan José Criado-Álvarez

**Affiliations:** 1Primary Care, Health Service (SESCAM), 13500 Puertollano, Ciudad Real, Spain; 2Department of Medical Sciences, Faculty of Health Sciences, Castilla-La Mancha University, 45600 Talavera de la Reina, Toledo, Spain; 3Integrated Attention Management of Talavera de la Reina, Castilla-La Mancha Health Service (SESCAM), 45600 Talavera de la Reina, Toledo, Spain; 4Primary Care, Health Service (SESCAM), 13005 Ciudad Real, Spain; 5Emergency Department, Health Service (SESCAM), 13500 Puertollano, Ciudad Real, Spain; 6Department of Gynecology and Obstetrics, Health Service (SESCAM), 13005 Ciudad Real, Spain; 7Coordinator of the Research, Teaching and Library Unit, Puertollano Hospital, Health Service (SESCAM), 13500 Puertollano, Ciudad Real, Spain; 8Department of Gynecology and Obstetrics, Health Service (SESCAM), 41007 Toledo, Spain; 9Institute of Health Sciences of Castilla-La Mancha, 45600 Talavera de la Reina, Toledo, Spain

**Keywords:** primary care, lung ultrasound, COVID-19, B-lines, pneumonia

## Abstract

Background: Currently, there are few studies that have analyzed the benefits of using lung ultrasound in the field of primary care, including in homes and nursing homes, for patients with suspected COVID-19 pneumonia and subsequent follow-ups. The aim of this study was to demonstrate that lung ultrasound is a useful technique for triaging these patients. Methods: An observational and retrospective study of individuals who presented with clinical suspicion of SARS-CoV-2 pneumonia was carried out during the months of March to June 2020 in Health Center number 2 of Ciudad Real and in homes of patients and nursing homes belonging to the Health Service of Castilla-La Mancha (Spain). Results: A total of 209 patients, of whom 86 (41.1%) were male, were included in the study. The most frequent ultrasound findings were bilateral B-lines, with a right predominance, specifically in the posterobasal region. Additionally, there was a statistical significance (*p* < 0.05) correlation between pathological positivity on lung ultrasound and PCR and chest X-ray positivity. When calculating the sensitivity and specificity of ultrasound and X-ray, ultrasound had a sensitivity of 93%, and X-ray had a sensitivity of 75%. Conclusion: Due to its high sensitivity and negative predictive value, lung ultrasound is very useful as a triage tool for patients with suspected SARS-CoV-2 pneumonia.

## 1. Introduction

The first cases of COVID-19 pneumonia occurred in December 2019 [1] in Wuhan, China. By the end of February, the number of those infected had increased significantly in the country, and by this date it had already spread to 28 other countries [2]. In Spain, a state of alarm due to the new coronavirus was declared on 14 March 2020 [3]. On 28 April, a plan was created that addressed four phases of de-escalation to increase socioeconomic activity and mobility through 21 June, the day that the state of alarm ended, i.e., marking the first day of “the new normal” and the resumption of previously restricted activities during the period of confinement [4].

The main risk groups that were affected were elderly individuals institutionalized in nursing homes as well as people with associated cardiovascular risk factors, such as obesity and immunosuppression. Additionally, health and nursing home personnel were at high risk of exposure [2,5].

The most common routes of transmission are similar to those for other respiratory viruses, such as through aerosols or through contaminated objects [6]. COVID-19 manifests as an acute respiratory infection, although asymptomatic or barely symptomatic cases have been widely reported, especially in the initial stages of the disease. The most common symptoms are fever, asthenia and dry cough. Dyspnea usually begins on the fourth or fifth day. Another associated clinical picture is headache, myalgia, abdominal pain, dizziness, nausea, diarrhea, vomiting, anosmia and ageusia [7].

The most commonly used type of diagnostic test is PCR (polymerase chain reaction), which is the reference standard and preferred technique for infection diagnoses, antigen detection tests and antibody detection tests (IgM and IgG) [8,9,10,11].

In COVID-19, the respiratory system is first affected; therefore, chest X-ray is commonly performed first. However, in many cases, the diagnostic performance of chest X-ray in the early stages of the disease is limited [12,13,14].

Lung ultrasound can be performed at the bedside without the need for patients to go anywhere, thus avoiding transferring patients suspected of infection, as recommended by the Spanish Society of Medical Radiology; it is also a fast, economical, simple and safe technique for patients [15]

At the beginning of the pandemic, because of the lack of knowledge about disease management and the saturation of hospital services, both in wards and in ICUs, the involvement of primary care was prioritized to regulate and channel the transfer of patients to hospitals and to actively collaborate in the management (diagnosis, classification and follow-up until discharge) of patients who could be treated on an outpatient basis [16].

In primary care and especially in epidemic situations, it is essential to have easily accessible, reliable techniques that can be performed anywhere. Ultrasound is a technique with great potential for respiratory pathology triage [17] because it is highly sensitive, even more sensitive than X-ray, with regard to visualizing consolidations, interstitial pathology and pleural effusion [18].

The objective of this study was to determine whether performing outpatient lung ultrasound is useful for triaging patients who come to a health center with symptoms of suspected SARS-CoV-2 pneumonia. In addition, we analyzed its sensitivity and specificity and determined whether, compared to chest X-ray, lung ultrasound could more accurately detect the presence of infiltrates.

## 2. Materials and Methods

### 2.1. Patients and Methods

An observational and retrospective study, conducted from March to June 2020, was carried out with patients at Health Center 2 of Ciudad Real (Centro de Salud número 2 de Ciudad Real) who presented with suspected symptoms of SARS-CoV-2 pneumonia, individuals in their homes who had difficulty moving to a hospital environment, either due to their baseline condition or due to the severity of their symptoms, and individuals in nursing homes in Ciudad Real, Calzada de Calatrava and Carrión, towns near the capital where the percentage of patients with suspected symptoms of pneumonia was high, all of them belonging to the Integrated Care Management Area of Ciudad Real of the Health Service of Castilla-La Mancha (SESCAM).

An organizational system was created in the health center for the classification of patients based on the clinical picture of the disease and lung ultrasound to establish adequate referral criteria (to the home or to the hospital).

The inclusion criteria were patients with clinical suspicion of pneumonia, such as fever greater than 38 °C for more than 2 days or low-grade fever for more than 5 days, dyspnea, musculoskeletal pain, oxygen saturation lower than 95% or mild symptoms but with significant comorbidities (diabetes, obesity, arterial hypertension, chronic kidney disease, heart disease, pulmonary pathologies, age over 65 years and immunosuppressive treatments), who had consulted by telephone with their doctor or had gone to the health center. Upon enrollment in the study, an assessment was conducted by the health center team to record symptoms and take vitals (oxygen saturation and temperature) to establish severity criteria. The exclusion criteria were patients with mild catarrhal symptoms less than 5 days, anosmia, ageusia, diarrhea, odynophagia, dysphonia, mild cough, no dyspnea or warning signs and oxygen saturation greater than 95%.

Patients with oxygen saturation greater than 95% and with mild symptoms, without fever or associated dyspnea, were treated with antibiotics or corticoisteroids from the Health Center, and they were followed-up by telephone every 48 h. Patients who were clinically unstable or stable but had bilateral confluent B-lines in both lung fields were referred to the hospital.

Patients who met the previously agreed upon inclusion criteria underwent a lung ultrasound, and for those with unilateral or rarely bilateral B-lines and who presented a good general condition, outpatient follow-up was performed according to the protocol then in effect [15], with telephone follow-up every 48 h. For those patients who had been prescribed hydroxychloroquine or who reported clinical worsening despite adhering to the prescribed treatment, a new ultrasound evaluation was performed after 1 week to assess the evolution.

The variables that were studied included sociodemographic data, location of the ultrasound examination (health center, nursing home or home), personal history (previous cardiac or pulmonary pathology, overweight or obesity and smoker), physical examination data (temperature or oxygen saturation), symptoms presented (fever, dyspnea, vomiting, nausea, diarrhea, cough, ageusia or anosmia), ultrasound findings (presence of pathological B-lines, pleural effusion and consolidations), location of the affected pulmonary fields, PCR test results, which were requested and performed by primary care units for mild–moderate cases and in hospitals for those patients referred to the hospital, and blood test or chest X-ray results.

The equipment used for the ultrasound studies was an Alpinion Encube^®^ and a Butterfly^®^ probe, both allow for the obtaining of quality images of the chest. The main difference between the 2 systems was that the probe for the Butterfly^®^ ultrasound did not need to be changed because it had a single silicon chip; therefore, through an electronic device such as a computer, mobile phone or tablet to view the images, it was only necessary to select the type of ultrasound to be performed, which in our case was the pulmonary mode [19]. All studies were performed with a convex probe.

The examination was performed in a sitting position when the patient’s baseline situation permitted and in a supine position for patients who were unable to sit up. Starting from superior to inferior, the hemithorax was divided into 6 zones (2 anterior, 2 lateral and 2 posterior), making a sweep in each of zone. In total, 12 areas were examined [18,20]. Depending on the experience of the examiner, the completion time varied between 5 and 10 min [18,21].

It should be mentioned that one operator was a professor of primary care ultrasound and both of them had accredited courses and years of training in ultrasound, using it as another tool in their daily clinical practice.

The ultrasound results were described based on the artifacts found. A partial loss of aeration due to occupation of the interstitial space (either by fluid or by fibrous tissue) resulted in the appearance of artifacts called B-lines [22]. B-lines are vertical artifacts resembling “comet tails”, which are hyperechoic, arising from the pleural line, and reach the end of the screen without attenuation. They move synchronously with pleural sliding [23]. According to the International Consensus [23], the presence of 3 or more B-lines in at least 2 thoracic areas in each hemithorax indicates diffuse interstitial lung syndrome. The total loss of aeration manifests as a consolidation pattern, as occurs, for example, in pneumonia or atelectasis. Some consolidations may present with associated pleural effusion in the supine position [23]. That is, a pathological ultrasound is defined by the presence of more than 3 B-lines per intercostal space or the presence of other findings, such as a consolidation pattern or pleural effusion.

### 2.2. Statistical Analysis

A descriptive analysis of the parameters was performed based on the scale of each variable. The Kolmogorov–Smirnov and Levene tests, with Lilliefor’s significance correction, were used to assess the distribution of the data for each variable. In the inferential analysis of independent variables, ANOVA was used to study the relationship between continuous variables and nominal variables or the relationship among n independent groups; if the outcome variable was dichotomous, Student’s t test was used. If both variables were dichotomous, the chi-squared test was used. A confidence level of 5% was established.

For the data analysis, the statistical program SPSS for Windows (Statistical Package for the Social Sciences version 28.0) was used.

### 2.3. Ethical Considerations

This study was approved by the Ethics and Drug Research Committee of the General University Hospital of Ciudad Real (Registry number: C-387) of the Integrated Care Management Area of Ciudad Real of the SESCAM and the Recommendations of the Bioethics Committee of Spain of April 2020 [24].

## 3. Results

A total of 209 patients were studied, of whom 86 (41.1%) were men and 123 (59.9%) were women. The mean age was 66.5 years (median: 67.0; SD: 18.66). The majority of the patients, 115 (55%), were treated in the health center, 73 (34.9%) were residents in nursing homes and 21 (10%) resided in their own homes. Eighty (38.3%) patients were referred to hospital care, while one hundred and twenty-nine (61.7%) remained at home. The PCR, ultrasound and X-ray data are provided in Table 1.

According to the main symptoms of the patients, they are included in the following Figure 1.

Regarding the lung ultrasound examination, some findings in Table 2 are notable. A total of 41.2% of the patients examined had bilateral B-lines, and 1% had bilateral pleural effusion.

Regarding the distribution of the ultrasound findings, 37 (26.4%) patients had right involvement, 20 (14.2%) had left involvement and 83 (59.2%) had bilateral involvement.

Of the total number of patients, 79 (37.7%) had a control ultrasound performed, among whom 37 (47%) had pathological findings. The most frequent findings in the control ultrasound were the presence of right unilateral B-lines (8%) and irregular bilateral B-lines with aerated areas (7.2%).

Twenty-one (10.3%) patients underwent CT, among whom fourteen (6.6%) had pathological findings.

Table 3 provides a comparison of pathological ultrasound results and PCR results.

There were statistically significant differences (*p* < 0.05) between the PCR and pathological ultrasound positivity. Ninety-three (92.1%) of the patients included in the study with suspected COVID-19 pneumonia with a positive PCR result had pathological findings on the lung ultrasound.

Statistically significant differences (*p* < 0.05) were also observed when analyzing whether there were differences between the pathological ultrasound findings and the pathological X-ray findings. Seventy-one (70.6%) patients with pathological ultrasound findings had pathological X-ray findings, and only 29.4% of those with pathological ultrasound findings had normal X-rays. No significant differences (*p* > 0.05) were observed for sex with regard to the following ultrasound patterns: confluent B-lines, right B-lines, left B-lines, bilateral lines or right and left condensations.

Of the one hundred and seventeen patients who had X-rays, eight patients (corresponding to 6.8% of the total X-rays and 10.3% of the total pathological X-rays) presented pathological findings on the lung ultrasound but normal X-rays, with a mean of 7 days until the appearance of lesions in the second X-ray (median: 4.5; SD: 6.88). It is noteworthy that of these eight patients, the majority were women and the pathological findings found were mostly bilateral B-lines followed by right subpleural involvement.

Regarding the risk factors studied, 102 (49.5%) of the patients were hypertensive, 32 (15.5%) were diabetic and 58 (28.2%) were dyslipidemic. Twelve patients (7.3%) were smokers, and forty-three (20.8%) had some type of heart disease. Statistically significant differences were found in terms of sex and pulmonary pathology and the presence of expectoration (*p* < 0.05). The most frequent symptoms in our study population were cough (46.6%), low-grade fever (36.2%) and dyspnea (30.6%). A total of 91.7% of the patients remained eupneic.

Sixty-eight (32.5%) patients were admitted to the ward, and one hundred and forty (67.3%) were not; three (1.4%) were admitted to the ICU. Twenty-one (10%) patients died, of whom nine (0.5%) died in the hospital itself, five (2.4%) died upon readmission days or weeks after the first hospital discharge or diagnosis, and seven (3.4%) died at home or in the nursing residence within the first 2 weeks after diagnosis.

There were no statistically significant differences (*p* > 0.05) between sex and age, admissions, discharges, control ultrasound and X-rays. No statistically significant differences were found between sex and referral, blood group, rh, overweight, obesity, hypertension, diabetes, dyslipidemia, smoking status or previous cardiac or pulmonary pathology.

When calculating the sensitivity and specificity of both tests in all patients, ultrasound had a sensitivity of 93% (95% CI: 87–97%), and X-ray had a sensitivity of 75% (95% CI: 65–85). For ultrasound, the specificity was 24.4% (95% CI: 11–38), with a positive predictive value of 75% (95% CI: 67–83) and a negative predictive value of 55.6% (95% CI: 33–79). For X-rays, the specificity was 41.4% (95% CI: 23–59), taking PCR as a reference, with a positive predictive value of 76% (95% CI: 66-86) and a negative predictive value of 40% (95% CI: 22–58).

In addition, taking into account the difference in days between performing the lung ultrasound and the PCR test, for tests performed less than 14 days between the two, the sensitivity increased to 96% (95% CI: 92–100), and the specificity increased to 28% (95% CI: 10–46); for tests performed less than 7 days between the two, the sensitivity was 98% (95% CI: 95–102), and the specificity was 26% (95% CI: 8–44).

The area under the curve (ROC) was less than 0.7 (95% CI: 0.333–0.667), decreasing to 0.5 (95% CI: 0.322–0.678) when only considering a difference of less than 7 days between the lung ultrasound and PCR (Table 4); the difference was not statistically significant (*p* > 0.05).

When analyzing the area under the curve for lung ultrasound with respect to X-rays and chest CT, the area under the curve for lung ultrasound presents was 0.5 (0.249–0.751), and that for X-rays was 0.453 (0.203–0.703); the results for both tests were lower than the CT, i.e., 0.603 (0.358–0.847), as shown in Figure 2.

## 4. Discussion

There are many studies that recommend lung ultrasound as an aid in the diagnosis and monitoring of patients with dyspnea [20,21,22,25] as well as in the triage of patients who consult a respiratory clinic [17,20,26,27,28], but we did not identify any previous study that evaluates the usefulness of lung ultrasound in primary care, including in the home and in nursing homes, for patients with clinical suspicion of SARS-CoV-2 pneumonia.

During the first wave of the COVID-19 pandemic, primary care was responsible for the assessment, triage and monitoring of patients who consulted health centers, both in person and by phone [29,30], and for the follow-up of hospital discharges. One of the barriers was the difficulty in providing chest X-rays, for which it is almost always necessary to move the patient [17], whether in the hospital or as an outpatient, entailing an avoidable added risk and in turn contributing to the worsening of the existing collapse of emergency services. Therefore, it was necessary to use lung ultrasound to help classify patients who began to present pulmonary involvement before serious complications appeared, becoming crucial in decision-making [25,26].

After analyzing the different variables, there were statistically significant differences between the pathological ultrasound findings and PCR positivity. There were also differences between the pathological ultrasound findings and the pathological chest X-ray findings. In addition, as mentioned in previous studies [17,21,26,31,32,33,34], lung ultrasound has proven to be highly sensitive (greater than 90%), which supports the hypothesis that performing an outpatient lung ultrasound can be useful for triaging patients on the basis of severity into mild, moderate and severe pneumonias and, therefore, for better patient control from the beginning, without subjecting patients to ionizing radiation, which is an enormous advantage for certain populations, such as pregnant women and children [16,33], and avoiding the need of going to the hospital for those who present clinical stability [20]. In this way, it was possible to monitor a large percentage of patients with mild–moderate COVID-19 pneumonia on an outpatient basis at the time of maximum hospital collapse.

However, the specificity of the test was limited [17,31,35]; therefore, the findings for patients with pulmonary pathology, such as the presence of B-lines, only indicated that there was pulmonary involvement, without discriminating its origin. Therefore, it is essential to address the clinical context of the patient because similar ultrasound findings can be found in patients with acute heart failure with pulmonary congestion or other pathologies, such as pulmonary fibrosis [20].

Importantly, lung ultrasound detected pulmonary involvement in a percentage of mild–moderate patients (8%) for whom the chest X-ray was normal or inconclusive, making it a potential tool to evaluate individuals in the early phases of COVID-19. In addition, the ROC curve analysis revealed that lung ultrasound, as a diagnostic modality, was not inferior to chest X-rays in patients with suspected SARS-CoV-2 pneumonia; however, as shown in other studies [20,36], chest CT continues to be the imaging test with the greatest diagnostic performance.

The most frequently found patterns were bilateral B-lines, with a right predominance, specifically in the posterobasal region, and in the control ultrasound, the most frequent findings were right unilateral B-lines and dispersed bilateral B-lines with aerated areas.

## 5. Conclusions

Lung ultrasound was useful as a triage tool for patients with suspected SARS-CoV-2 pneumonia. Due to its high sensitivity and negative predictive value, the existence of pulmonary involvement could be ruled out with great accuracy in patients who had clinical suspicion of infection by the novel coronavirus; furthermore, lung ultrasound allowed adequate follow-up for those patients who present pathological findings on the ultrasound and early referral to a hospital service for those patients who, despite not presenting clinical warning signs, presented a pattern of bilateral pulmonary involvement in an initial ultrasound. All this could be done in the first patient assessment, regardless of where the patient was.

## 6. Limitations of the Study

Among the limitations of the study was the sample size because only ultrasounds performed during the first wave of the pandemic that met the aforementioned inclusion criteria, performed during the months of March to June 2020, were selected. Lung ultrasound, although having a steep learning curve [16,21], is an operator-dependent technique [16,31,37]; therefore, the interpretation of the results depended on the training and experience of the clinician who performed it. Third, the ultrasound examinations were performed at a time of maximum health system collapse, when the option to perform PCR on all patients in the study was not possible; therefore, not all ultrasound scans could be compared with the diagnostic standard, i.e., PCR test results. Fourth, the ultrasounds that were performed by each team were not recorded; therefore, it was not possible to analyze the differences between the variability of the different ultrasound machines. Fifth, ultrasound is a technique with great sensitivity but low specificity [17,21,31,35]; therefore, it is difficult to distinguish whether lung injuries are caused by other pathologies, and the high prevalence of respiratory symptoms attributed to COVID-19 may generate a bias [34].

## Figures and Tables

**Figure 1 jcm-11-06420-f001:**
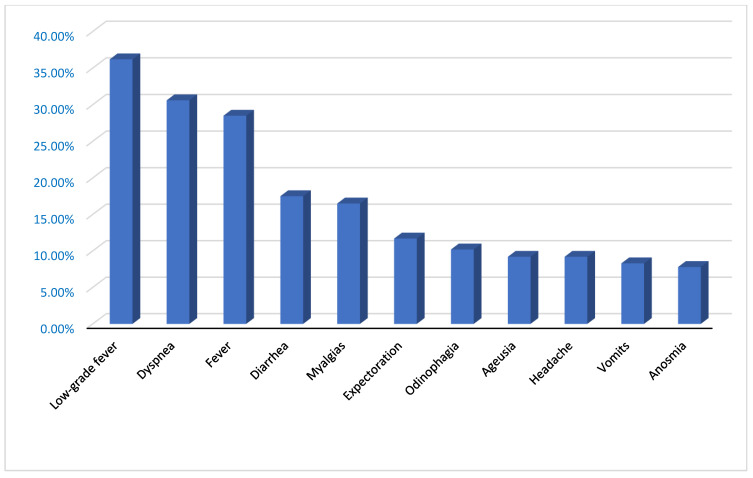
Main symptoms of patients.

**Figure 2 jcm-11-06420-f002:**
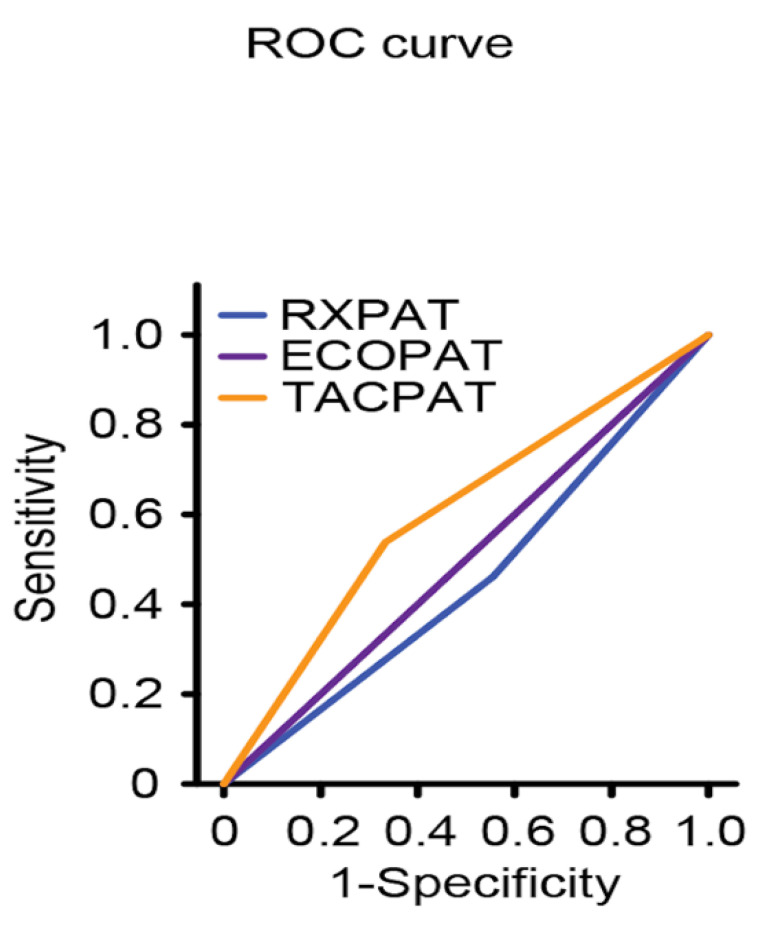
Comparison of the areas under the curve for pathological lung ultrasound, pathological chest CT and pathological chest X-ray.

**Table 1 jcm-11-06420-t001:** Diagnostic test results.

	Pathological or Positive	Normal or Negative
PCR (n: 142)	101 (71%)	41 (29%)
Ultrasound (n: 209)	167 (80%)	42 (20%)
X-ray (n: 117)	77 (65.8%)	40 (34.2%)

n: number of subjects. PCR: Quantitative polymerase chain reaction.

**Table 2 jcm-11-06420-t002:** Analysis of lung ultrasound characteristics.

	Yes		No		n Total
	n	%	n	%	
Confluent B-lines	27	13.3	176	86.7	203
Right unilateral B-lines	37	18.1	167	81.9	204
Left unilateral B-lines	22	10.8	182	89.2	204
Bilateral B-lines	84	41.2	120	58.8	204
Right subpleural consolidation	7	3.4	197	96.6	204
Left subpleural consolidation	2	1.0	202	99.0	204
Bilateral subpleural consolidation	3	1.5	201	98.5	204
Right pulmonary condensation	14	6.9	190	93.1	204
Left pulmonary condensation	8	3.9	196	961	204
Bilateral pulmonary condensation	3	1.5	201	98.5	204
Right pleural effusion	3	1.5	201	98.5	204
Left pleural effusion	4	2.0	201	98.5	205
Bilateral pleural effusion	2	1.0	202	99.0	204

n: number of subjects.

**Table 3 jcm-11-06420-t003:** Pathological ultrasound-PCR relationship.

		Pathological PCR		
		YES	NO	Total
Pathological Ultrasound	YES	93 (75%)	31 (25%)	124 (100%)
	NO	8 (44.4%)	10 (55.6%)	18 (100%)
	Total	101 (71.1%)	41 (28.9%)	142 (100%)

**Table 4 jcm-11-06420-t004:** Sensitivity, specificity, area under the curve, positive predictive value and negative predictive value as a function of time between PCR and ultrasound.

	Time between PCR and Ultrasound		
	7–14 Days n = 102 (48%)	Less than 7 Days n = 83 (39.7%)	Less than 5 Daysn = 72 (34.4%)
**Sensitivity (%)**	96 (95% CI: 92–100)	98 (95% CI: 95–102)	98 (95% CI: 94–102)
**Specificity (%)**	28 (95% CI 10–46)	26 (95% CI: 8–44)	29(95% CI: 9–48)
**Area under the ROC curve**	0.471 (95% CI: 0.308–0.633)	0.5 (95% CI: 0.333–0.667)	0.5 (95% CI: 0.322–0.678)
**Positive predictive value (%)**	80 (95% CI: 72–89)	78 (95% CI: 68–87)	77 (95% CI: 67–87)
**Negative predictive value (%)**	70 (95% CI: 42–98%)	86 (95% CI: 60–112)	86 (95% CI: 60–112)

n: number of subjects. ROC: Receiver Operating Characteristic.

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
