# Peer review of "Lung Ultrasound as a Triage Method in Primary Care for Patients with Suspected SARS-CoV-2 Pneumonia"

_jcm, 2022, doi:10.3390/jcm11216420_

Round 1

Reviewer 1 Report

This paper introduces a new technique for detecting COVID-19 pneumonia. It describes many measurement indicators in depth, points out the shortcomings of existing technologies, and also describes the characteristics of lung ultrasound with high sensitivity and negative predictive value. No matter where the patient is, the evaluation can be completed in the initial ultrasound examination.

First of all, this article is well written. The author clearly points out the shortcomings of the existing technology, points out the problems that can be solved by the new technology in this article, and finally points out the existing limitations of this technology and the areas that need to be improved. The writing structure is also logical. The paper may be consider to be accepted if the following concerns could be well addressed. 

1. there are few materials about the detection methods in the article. Please describe them in more detail.

2. it will be easier to understand if the authors add charts to the statistical analysis.

3. English writing should be improved by native speakers. 

Reviewer 2 Report

The manuscript's topic represents a really interesting theme in the Covid-19 management and diagnosis and is well written and clearly exposed;

I've only three minor comments/suggestions for Authors:

1. since that's a very important concern, could you report more details about the 8 patients with pathological US and normal x-ray (i.e., clinical data, Covid-19 outcome, time to x-ray positivity, ..) ?

2. are data available on operators experience on lung US? since this is a highly operator-dependent procedure, it could be of interest for readers to know the operators' expertise

3. please, try to update the reference on the topic (some interesting published data are available to date)

Round 2

Reviewer 1 Report

The revised manuscript has addressed the reviewer's concerns.